# Analysis of the Fruit Drop Rate Caused by Typhoons Using Meteorological Data

Su-Hoon Choi [1,†], So-Yeon Park [1,†], Ung Yang [2], Beomseon Lee [3], Min-Soo Kim [4] and Sang-Hyun Lee [2,5,6,*]

1    Department of Mathematics and Statistics, Chonnam National University, Gwangju 61186, Republic of Korea;
     magafand@naver.com (S.-H.C.); syeonon1512@gmail.com (S.-Y.P.)
2    Asian Pear Research Institute, Chonnam National University, Gwangju 61186, Republic of Korea;
     norcardia78@gmail.com
3    Industry-Academic Cooperation Foundation, Sunchon National University,
     Sunchon 57922, Republic of Korea; bslee@scnu.ac.kr
4    Department of Statistics, Chonnam National University, Gwangju 61186, Republic of Korea;
     kimms@chonnam.ac.kr
5    Department of Horticulture, College of Agriculture and Life Sciences, Chonnam National University,
     Gwangju 61186, Republic of Korea
6    Interdisciplinary Program in IT-Bio Convergence System, Chonnam National University,
     Gwangju 61186, Republic of Korea
*    Correspondence: pear@chonnam.ac.kr
†    These authors contributed equally to this work.

**Abstract:** Typhoons, which are a common natural disaster in Korea, have seen a rapid increase in annual economic losses over the past decade. The objective of this study was to utilize historical crop insurance records to predict fruit drop rates caused by typhoons from 2016 to 2021. A total of 1848 datasets for the fruit drop rate were generated based on the impact of 24 typhoons on 77 cities with typhoon damage histories. Three different types of measures—the average value, the maximum or minimum value, and the value at a specific point during the typhoon—were applied to four meteorological factors, yielding a total of twelve variables used as model inputs. The predictive performance of the proposed models was compared using five evaluation metrics, and SHAP analysis was employed to assess the contribution of predictor variables to the model output. The most significant variable in explaining the vulnerability to typhoons was found to be the maximum wind speed. The categorical boosting model outperformed the other models in all evaluation metrics, except for the mean absolute error. The proposed model will assist in estimating the potential crop loss caused by typhoons, thereby aiding in the establishment of mitigation strategies for the main crop-producing areas.

**Keywords:** crop insurance data; fruit drop rate; machine learning; typhoon

## 1. Introduction

According to the United Nations (UN) global assessment report on natural disasters, both the extent of economic losses caused by natural disasters and the number of affected people are rapidly increasing [1]. The World Meteorological Organization (WMO) reported that a total of 11,072 natural disasters occurred globally from 1970 to 2019, of which 711 occurred in the 1970s, compared to 3536 in the 2000s. In particular, typhoons have caused significant damage, affecting 39% of individuals and contributing to 54% of property damage.

Thermal energy from the sun is the primary cause of changes in Earth's weather, and because the Earth is spherical, an imbalance in thermal energy exists between low and high latitudes. Convective clouds form in the sea near the equator, where the sun's elevation angle is high, resulting in the accumulation of a significant amount of energy. These clouds gather and develop into a massive low-pressure system, ultimately evolving into a typhoon.

The typhoon receives evaporated vapor from the sea and maintains its strength as it moves to high latitudes. Among tropical cyclones, the WMO classifies a typhoon as having a maximum wind speed of 33 m/s or more near the center, a strong tropical storm (STS) with 25 to 32 m/s, and a tropical storm (TS) with 17 to 24 m/s. However, in Korea, any tropical cyclone with a maximum wind speed of 17 m/s or more is called a typhoon [2].

The intensity of typhoons has increased in recent decades. According to the National Typhoon Center, "very strong" typhoons with a maximum wind speed of over 44 m/s accounted for half of the typhoons that hit South Korea over the 10-year period from 2009 to 2018. In addition, the annual number of "super strong" typhoons, with a maximum wind speed of 54 m/s or more, has increased from less than two in the 1990s and early 2000s to three or four, and even as many as seven in recent years. Typhoons mainly affect Korea between July and October, causing damages such as flooding, landslides, damage to agricultural facilities, reduced crop yields, and the deterioration of farm quality. Typhoon Rusa, which occurred in August 2002, resulted in the highest daily precipitation recorded in Gangneung-si, reaching 870.5 mm. This extreme weather event caused extensive damage, estimated at approximately 5.2 trillion won (KRW). In addition, Typhoon Maemi in September 2003 caused massive property damage, estimated at approximately 4.2 trillion won (KRW). The summer of 2003 had the worst harvest on record due to heavy rain, frequent abnormally low temperatures, and typhoon damage.

Research related to the impacts of typhoons on agriculture is crucial because typhoons not only affect the supply and demand for crops, but also impact the incomes of farmers. The authors of [3] attempted to estimate the economic losses caused by typhoons that occurred in Zhejiang Province, China from 1971 to 2008 by analyzing the occurrence of typhoons and environmental factors. An artificial neural network, an algorithm inspired by neural networks found in biology, was utilized to account for the intricate nonlinear relationship between evaluation factors and typhoon damage. A self-organizing radial basis neural network model was used in study [4] to predict the economic losses in the rice industry in Taiwan caused by typhoons. Typhoon characteristics and meteorological data were utilized to select the optimal network, resulting in prediction results that closely matched the actual losses. Another study established a loss assessment model for crop losses as well as the number of deaths, disappearances, and destroyed houses due to typhoons in Taiwan from 1965 to 2004 [5]. The model accurately estimates the expected losses by incorporating factors such as the maximum daily rainfall, the maximum central wind speed, the lowest central pressure, the radius of class-seven winds, and the typhoon-affected period. The impact of typhoons on agricultural losses in Taiwan from 2006 to 2019 has also been investigated, including the role of flood protection infrastructure [6]. The occurrence of natural disasters was found to have a high correlation with agricultural production. Extreme rainfall was found to cause the greatest agricultural damage, whereas there was a spatial difference in the agricultural losses caused by typhoons across various regions. The study [7] analyzed the impact of typhoons on rice loss in Philippine provinces. The ratio of the damaged area to the total area was used to determine rice loss. Additionally, a total of 14 geographical and typhoon characteristics were considered as explanatory variables.

Providing predictive results before the typhoon arrives is important as it allows for the establishment of preventive measures in response to the typhoon. In addition, due to the intensity, path, and regional characteristics of typhoons, detailed research is required on the degree of damage, as these factors can lead to variations in damage between adjacent areas. However, collecting detailed data on typhoon damage is difficult from both human and economic perspectives. Since 2001, crop insurance for apples and pears has been implemented in South Korea. This serves as a highly reliable source of data because it confirms the extent of damage for each farm, and compensation is only provided after a fair and objective investigation and evaluation of the damages. However, apart from a few prior studies, limited research has been undertaken on the fruit drop rate using crop insurance, such as the ones conducted by the authors of the present study [8,9]. The characteristics of

typhoons (e.g., the path, central atmospheric pressure, maximum wind speed, etc.) were used in [8] to predict the fruit drop rate of astringent persimmon caused by typhoons from 2016 to 2019. AutoML in the H2O library was utilized for this prediction, and the root-mean-square error (RMSE) and mean absolute error (MAE) were used to compare the predictive performance of multiple models. Extreme gradient boosting (XGBoost) was found to produce the best prediction performance of these models. The drop rate of pears caused by typhoons from 2016 to 2021 was studied in [9]. The study compared the distance from the center of a typhoon to the affected area with the strong wind radius in order to determine the time of impact during the typhoon. Additionally, the study presented methods for selecting representative values for various typhoon characteristics.

Despite previous research using crop insurance data to determine the fruit drop rate caused by typhoons; determining the susceptibility of the main crop-producing areas based solely on typhoon attributes is limited because the degree of their impact varies greatly between regions. To overcome this limitation, the present study analyzed the fruit drop rate due to typhoons according to region by utilizing meteorological information from weather stations installed across the country. When utilizing meteorological information, determining the most suitable representative measure is crucial in explaining the fruit drop rate. Therefore, three representative measures were selected for testing meteorological data: the average value; the maximum or minimum value; and the value at a specific time point during the progression of the typhoon.

The PyCaret library was used to compare different machine learning models for predicting fruit drop rates by region. It is possible to compare several models to identify meteorological factors that have a strong impact on the fruit drop rate and to determine the expected fruit drop rate based on available meteorological information across the country.

## 2. Materials and Methods

### 2.1. Materials

#### 2.1.1. Crop Insurance Data

Actual data on the fruit drop rate were calculated using historical damage records for crop insurance provided by Nonghyup Property and Casualty Insurance Co., Ltd. (Seoul, Republic of Korea), in cooperation with the Ministry of Agriculture, Food and Rural Affairs in Korea. Insurance data from apple orchards were utilized due to apples accounting for the largest cultivation area among fruit crops in Korea, and the fact that their harvest season coincides with the period when typhoons most commonly make landfall on the Korean Peninsula. Between 2016 and 2021, a total of 89,007 cases related to typhoon damage were examined across different regions, considering historical damage records. The crop insurance records list the types of fruit, the date of the damage, the total number of dropped fruits, and the total number of set fruits. The drop rate of apples is calculated by dividing the total number of dropped apples by the total number of set apples in the area.

#### 2.1.2. Typhoon Data

Among a total of 155 typhoons that occurred from 2016 to 2021, information on the path (i.e., latitude and longitude) and the radii of strong winds for 24 typhoons that affected the Korean Peninsula (referred to hereafter as impact typhoons) were collected from the Korea Meteorological Administration (KMA) database. Additionally, based on the Disaster Annual Report produced by the Korean Ministry of the Interior and Safety, attribute values for the typhoons were collected only for the period during which typhoon damage occurred on land rather than the period from occurrence to extinction of the typhoon (Table A1).

#### 2.1.3. Meteorological Data

The KMA operates the Automated Synoptic Observing System (ASOS) and Automatic Weather Station (AWS) to provide meteorological information. The ASOS detects local meteorological phenomena through automatic and visual observations and is currently installed at 103 meteorological offices across the country. The AWS is installed at 510 observatories

nationwide, encompassing major observation points and mountainous areas, where basic atmospheric variables are automatically measured.

After typhoon landfall, vulnerability to typhoons, which varies significantly by region, may be better explained by utilizing meteorological data from specific areas affected by typhoons at a specific time point rather than relying on the overall attribute values of typhoons. Meteorological data were collected from the ASOS and AWS, which are close to the observed typhoon's location at a specific time point, in order to provide a more accurate representation of the meteorological conditions in those areas. Typhoons are known to have a close relationship with both wind speed and precipitation [10]. The strong winds can cause water stress due to forced transpiration, stripping, and injuring plant organs. Continuous flooding caused by excessive precipitation can lead to decreased photosynthesis and respiration [11]. Atmospheric pressure and air temperature were also considered. The study [12] illustrates the inverse relationship between surface pressure and sustained wind speed for numerous tropical cyclones. The average temperature and maximum temperature exhibited a negative correlation with typhoon damage and flooded area [13]. Additionally, an increase in temperature reduces the amount of moisture present in the atmosphere, leading to a decrease in relative humidity [14].

### 2.2. Methods

### 2.2.1. Model Selection and Description

This study used PyCaret for regression and prediction analysis. PyCaret is an efficient end-to-end machine learning and model management tool with a quick learning time. The low-code library is easy to use for running models and is highly productive, thanks to the inclusion of several frameworks [15]. PyCaret can run a variety of models for classification, regression, clustering, and natural language processing. By optimizing evaluation scores through hyperparameter tuning and ensemble techniques, it can also enhance the performance of these models [16]. It also has the advantage of being able to run several models at the same time and compare their performances in a table.

Random forest (RF), extra tree (ET), gradient boosting machine (GBM), light gradient boosting machine (LightGBM), extreme gradient boosting (XGBoost), adaptive boosting (AdaBoost), and categorical boosting (CatBoost) were selected as representative machine learning models, and their results were compared using the mean absolute error (MAE), mean squared error (MSE), root-mean-squared error (RMSE), root-mean-squared log error (RMSLE), and coefficient of determination ($R^2$) as the evaluation metrics.

### Random Forest (RF)

RF is a non-parametric ensemble model that generates individual tree predictors by creating randomly independent bootstrap samples from the entire training dataset [17,18]. At each node of all trees, the best split is adopted from a subset of randomly selected input variables. Random selection of input variables reduces the correlation between unpruned trees and suppresses bias [19]. The prediction of each tree being independent, along with these characteristics, limits the error when generalizing to a large number of trees, preventing over-fitting [20]. The final predicted value of the RF model is determined by taking the average of the predicted values from $k$ individual trees. This approach enables the generation of smooth functions that help reduce sample variance [21].

### Extra Tree (ET)

ET, based on RF, is a tree-based model. RF divides nodes through an optimization process, while ET trains predictors after a node split using randomly selected features with all cutoffs. The bias in ET can be reduced by learning using all of the data, unlike in RF where bootstrap samples are used [22]. One advantage of random splitting is its low risk of over-fitting and reduced variance.

Adaptive Boosting (AdaBoost)

AdaBoost is a sequential ensemble model that combines several weak learners randomly to create a powerful learner [23]. In AdaBoost, weights indicate the relative importance of an instance and are used to calculate errors for the data. False predictions are highly weighted and the weights are updated with the corrected errors and predictor reliability, which is recalculated after each iteration [24]. The greater the loss, the greater the weight, which in turn increases the probability of selecting the corresponding instance to train subsequent primary learners. By repeating this process, the final output provides ensemble predictions by calculating the weighted median values for the individual models [25].

Gradient Boosting Machine (GBM)

GBM is a method that sequentially adds and combines new models to reduce residuals and improve the accuracy of the resulting estimates. The negative slope of the loss function associated with the entire ensemble is consistent with the goal of reducing the loss function, thus ensuring a high correlation with a new learner:

$$-g_m(x_i) = - \left[ \frac{\partial L(y_i, F(x_i))}{\partial F(x_i)} \right]_{F(x)=F_{m-1}(x)}$$

If the model is created sequentially through fitting the gradient to the residuals, the residuals will gradually decrease, ultimately leading to the development of a predictive model that effectively explains the training set [26]. In regression, optimization is mainly performed using the mean squared error as the loss function [27]. However, GBM can experience over-fitting where the use of too many iterations complicates the model and causes it to closely fit the training data, ultimately increasing the expected loss.

Extreme Gradient Boosting (XGBoost)

XGBoost is a technique developed from the existing GBM model that not only offers high predictive performance, but also demonstrates high learning and search speeds, thanks to its parallel and distributed computing process [28]. The loss function regulates the prediction performance based on the error, while the normalization function reduces complexity and over-fitting [29].

Light Gradient Boosting Machine (LightGBM)

LightGBM is a model that utilizes gradient-based one-side sampling (GOSS) and exclusive feature bundling (EFB) to reduce computational complexity and the time required to obtain information from all possible split points for high-dimensional data [30]. GOSS reduces the sample size by retaining data instances with large absolute slope values and randomly removing data instances with small slopes. This approach allows for obtaining more information compared to simply extracting data without considering the slope. EFB groups mutually exclusive variables together to reduce the number of variables. The feature space of high-dimensional data is very sparse, and the features within that space are mutually exclusive, which accelerates the training process without reducing accuracy.

LightGBM uses a leafwise partitioning scheme to split nodes, effectively creating a deep asymmetric tree. It continuously segments nodes with the maximum loss and the highest information gain. Despite its ability to reduce computational costs [31], it should be noted that LightGBM can still lead to over-fitting.

Categorical Boosting (CatBoost)

GBM is susceptible to over-fitting due to biased gradient estimates, while the prediction model obtained after several boosting stages exhibits a target leakage problem caused by shifting not only the learning sample, but also the test sample. CatBoost was designed to solve these problems [32]. $M_j$, a model employed in CatBoost, learns through examples with an order of up to the $j$-th random permutation with models $M_1, \cdots, M_n$.

To calculate the residuals from the $j$-th sample, the unbiased gradient of $X_j$ is estimated using the trained model $M_{j-1}$ without gradient estimation for the $j$-th sample. Then, the residuals are calculated from the predictions. The ordered boosting method prevents target leakage during sequential learning of the trees.

$$r^t(x_j, y_j) = y_j - M_{j-1}^{t-1}(x_j)$$

Random sampling of permutations during the learning process, for use in multi-model training, helps prevent over-fitting and reduces variance [33].

### 2.2.2. Evaluation Metrics

The use of a single evaluation metric is generally limited in predicting learning model errors as it emphasizes only specific aspects of the error characteristics [34]. Thus, in the present study, the MAE, MSE, RMSE, RMSLE, and $R^2$ were calculated to evaluate the performance of the tested models.

#### Mean Absolute Error (MAE)

The MAE is the average of the absolute value of the error, which is the difference between the actual value and the predicted value. The predicted value and the unit of error are the same, and all errors are equally weighted, making them robust even in the presence of outliers. If the importance of outliers is not high and a close-fitting model is targeted, it is an appropriate indicator.

$$\mathrm{MAE} = \frac{1}{N} \sum_{i=1}^{N} |y_i - \hat{y}_i|$$

#### Mean Squared Error (MSE)

Unlike the MAE, the larger the error, the more weight it is given, so this metric is sensitive to outliers. Avoiding under-predictions is important when dealing with outliers.

$$\mathrm{MSE} = \frac{1}{N} \sum_{i=1}^{N} (\hat{y}_i - y_i)^2$$

#### Root-Mean-Squared Error (RMSE)

The RMSE is less sensitive to outliers than MAE and more sensitive than MSE when applying routes. As with the MAE, the predicted value and the unit for the error are the same. The MAE, MSE, and RMSE depend on the scale, so if the size of the error is the same for each model, the error ratio may not be the same. The RMSLE is the most useful metric when the relative ratio of the error is important.

$$\mathrm{RMSE} = \sqrt{\mathrm{MSE}} = \sqrt{\frac{1}{N} \sum_{i=1}^{N} (\hat{y}_i - y_i)^2}$$

#### Root-Mean-Squared Log Error (RMSLE)

The RMSLE measures the relative error between the actual value and the predicted value using a ratio. It is more robust to outliers than the RMSE because it assigns a log function to the actual and predicted values. The RMSLE imposes a higher penalty by using the logarithm function when the predicted value is lower than the actual value. Therefore, the model tends toward over-prediction because it imposes a greater penalty for underestimation.

$$\mathrm{RMSLE} = \sqrt{\frac{1}{N} \sum_{i=1}^{N} (\log(y_i + 1) - \log(\hat{y}_i + 1))^2}$$

The MAE, MSE, RMSE, and RMSLE commonly range between 0 and $+\infty$, where a higher value signifies worse performance of the model, with 0 representing a perfect fit. However, the use of these metrics, each of which returns a single value to represent model performance, does not provide sufficient information on the regression performance in relation to the distribution of the actual values.

Coefficient of Determination ($R^2$)

In contrast to the other metrics, $R^2$ has a higher score when the predicted values are closer to the actual values [35]. $R^2$ is calculated as a function of the total variation of the actual values, considering the regression line and the mean. It indicates how close the values predicted by the model are to the actual values; scores closer to 1 represent a better performance.

$$R^2 = 1 - \frac{\sum_{i=1}^{N}(y_i - \hat{y}_i)^2}{\sum_{i=1}^{N}(y_i - \overline{y}_i)^2}$$

2.2.3. Shapley Additive Explanations (SHAP)

Understanding the influential factors and effects involved is crucial for accurately interpreting the predictive outcomes of models. Analyzing the significance of individual features offers insights into potential enhancements of the model and aids in understanding the modeling process, ultimately improving the reliability of the results. However, interpreting complex ensemble and machine learning models is usually challenging.

To address this issue, SHAP has been designed to assign importance values to specific predictions for each feature. Additive feature attributes are utilized to approximate an interpretable and simple explanatory model in order to accurately determine SHAP values. The explanatory model is a linear function of the binary variable $z' \in \{0, 1\}^M$, where $M$ represents the number of simplified input features:

$$g(z') = \varphi_0 + \sum_{i=1}^{M} \varphi_i z_i'.$$

The expansion model based on SHAP values is the only model that satisfies the three properties of additive feature attributes: local accuracy, missingness, and consistency [36]:

$$\varphi_i(f, x) = \sum_{z' \subseteq x'} \frac{|z'|!(M - |z'| - 1)!}{M!} \left[ f_x(z') - f_x(z'\backslash i) \right],$$

$$f_x(z') = f(h_x(z')) = E[f(z)|z_S] \approx f([z_S, E[z_{\overline{S}}]]),$$

where $|z'|$ is the number of non-zero entries in $z'$, and $z'\backslash i$ denotes that $z_i' = 0$. The contribution $\varphi_i$ of each feature to the model output in SHAP is assigned according to their marginal contribution [37,38]. Therefore, the SHAP value is directly proportional to the importance of the model's variables, allowing for the visualization and weighting of the most significant variables.

**3. Analysis Framework**

Figure 1 displays an overview of the steps in the analysis framework.

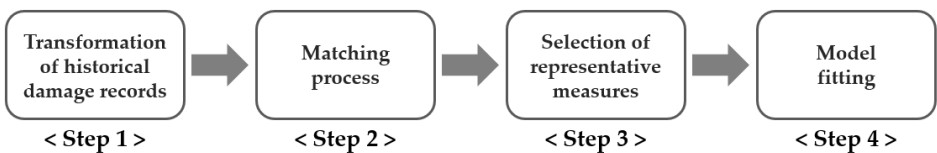

**Figure 1.** Overview of the steps in the analysis framework for the present study.

### 3.1. Transformation of Historical Damage Records from the Orchard to the City Level

Meteorological information was utilized in the present study to determine the impact of typhoons in various regions. Because meteorological observations from weather stations were only available at the city level, crop insurance data were also combined at the city level, even though historical insurance claim records were based on individual orchards.

The fruit drop rate was calculated by dividing the total number of dropped apples by the total number of apples set. Data were collected from 77 cities that had apple orchards with historical damage records for the period of 2016–2021, resulting in a total of 1848 datasets across the 24 impact typhoons that occurred in Korea during this period. If cities did not have historical damage records for a particular typhoon, the fruit drop rate would be set to 0.

### 3.2. Matching Process between Cities and Weather Stations

The closest meteorological station to the city with historical damage records was determined to associate the impact of individual typhoons with meteorological data. During the analysis period, meteorological data from the ASOS and AWS were utilized, and weather stations with missing values were excluded from the matching process. Four meteorological factors related to typhoon (i.e., air temperature, atmospheric pressure, precipitation, and wind speed) were used as explanatory variables, as mentioned in Section 2.1.3. After performing exploratory data analysis, outliers such as negative values in atmospheric pressure data were removed. Additionally, linear interpolation was utilized to fill in missing values for meteorological data during the typhoon period. This was conducted by extrapolating from the previous and subsequent time points.

### 3.3. Selection of Representative Measures for the Meteorological Variables

Various meteorological data were available for the period when a typhoon affected the main apple-producing areas. However, the fruit drop rates were calculated based on the cumulative damage after the typhoon had passed. Therefore, it was necessary to determine a representative measure from the available meteorological data that could reliably reflect the fruit drop damage while constructing a prediction model based on meteorological data for fruit drop rates.

The representative measure employs three approaches to differentiate the cumulative, instantaneous, and time point effects of each meteorological variable:

- Approach 1: The use of the average value for the meteorological variables.
- Approach 2: The use of the maximum or minimum value for the meteorological variables.
- Approach 3: The use of the value for the meteorological variables at a specific time point during the typhoon progression.

The average is commonly used to identify cumulative effects over a period. However, since it is difficult to show instantaneous effects with the average, an approach using the maximum or minimum value as a representative measure was introduced. Each meteorological variable was determined in the direction of increasing damage. The maximum precipitation and wind speed, as well as the minimum air temperature and atmospheric pressure were utilized. The third approach involved utilizing meteorological data at a specific time point when the fruit-producing area was affected by the typhoon. This is because it was believed that among the different time points of the typhoon, there might be a specific point in time that had the most significant impact on the affected area. For example, collecting meteorological data at the point when the strong wind radius was the largest or when the typhoon was the closest to the affected area may more accurately reflect vulnerability to a typhoon. To calculate the lowest value, subtract the radius of the strong wind from the distance between the typhoon center and the main producing area:

$$\text{argmin}(D_{i,t} - R_{i,t}),$$

where $D_{i,t}$ and $R_{i,t}$ represent the distance from the typhoon center to the affected area and the radius of strong wind, respectively, at the *t*-th time point of the *i*-th typhoon.

A total of twelve variables were employed as input variables, including the four meteorological variables used to explain typhoon damage. These three approaches were applied for the selection of representative values.

### 3.4. Model Fitting

The 1848 datasets were randomly divided into training and test sets in a 6:4 ratio, and the model architecture was trained using 5-fold cross-validation. All analyses were conducted using Python (version 3.8.5) and the PyCaret library (version 2.3.10) [39]. To assess the impact of predictor variables in the proposed model, we utilized the TreeExplainer from the SHAP library (version 0.40.0) [40,41].

## 4. Results

### 4.1. Exploratory Data Analysis

The mean and standard deviation of the twelve explanatory variables and the one dependent variable used for model fitting are summarized in Table 1. An analysis of variance (ANOVA) was conducted to determine whether there were differences in the average values of the three representative measures used for each of the four meteorological factors. The ANOVA confirmed that there was a difference in means among the three approaches ($p < 0.001$).

**Table 1.** Summary of all variables [1].

|  | Variables | Selection Criteria for the Representative Measure | | | *p*-Value [2] |
|---|---|---|---|---|---|
|  |  | Approach 1 | Approach 2 | Approach 3 |  |
| Input | Air temperature | 23.61 (3.90) | 18.78 (4.70) | 23.39 (3.83) | <0.001 |
|  | Atmospheric pressure | 994.60 (12.76) | 985.57 (14.53) | 987.33 (14.83) | <0.001 |
|  | Precipitation | 0.64 (0.73) | 9.58 (10.32) | 2.27 (5.41) | <0.001 |
|  | Wind speed | 1.90 (0.79) | 5.42 (2.21) | 3.19 (2.39) | <0.001 |
| Output | Fruit drop rate | 0.02 (0.06) | | | - |

[1] Mean (standard deviation). [2] As determined using ANOVA.

### 4.2. Variable Importance in the Proposed Model Based on SHAP Values

A SHAP analysis was conducted, using 5-fold cross-validation, to assess the contribution of predictor variables to the output of the optimized model. The SHAP values determine the impact of each variable on the model output by comparing the extent to which the absence or presence of a particular variable changes the output of the model.

The importance of the maximum wind speed (Approach 2) in terms of the model output was highlighted via the SHAP analysis, as indicated in Table 2, as it resulted in the highest mean absolute SHAP value among all models. Although there were minor differences between the proposed models, the wind speed and atmospheric pressure using approaches 2 and 3, respectively, were found to be the most important meteorological variables in explaining fruit drop rates.

### 4.3. Differences in Selection Criteria for Representative Values for Each Meteorological Factor

A total of twelve input variables were utilized in the models, resulting from the combination of three selection criteria and four meteorological factors. The feature importance analysis showed that the variables having the greatest impact on the model output differed in terms of the measure used to represent the data. Therefore, the approach with the highest SHAP value was investigated to determine the optimal selection criteria for each meteorological factor (Table 3). For most meteorological factors except wind speed, the representative values derived from Approach 3 produced the highest SHAP values in the



proposed models. Approach 2, using the maximum value as the model input, provides the best explanation for vulnerability to typhoons in terms of wind speed. These results indicate that the generalization of the fruit drop rate prediction model can be facilitated by using extreme values or values at a specific time point, rather than average values, as model inputs.

**Table 2.** Variable importance based on mean absolute SHAP value.

| Model [1] | Variable Importance Ranking [2,3] | | | | |
|---|---|---|---|---|---|
| | 1 | 2 | 3 | 4 | 5 |
| RF | Wind speed (A2) 0.008 | Atmospheric pressure (A3) 0.006 | Precipitation (A3) 0.005 | Atmospheric pressure (A2) 0.004 | Wind speed (A3) 0.004 |
| ET | Wind speed (A2) 0.013 | Atmospheric pressure (A2) 0.005 | Wind speed (A3) 0.005 | Atmospheric pressure (A3) 0.004 | Precipitation (A3) 0.004 |
| AdaBoost | Wind speed (A2) 0.019 | Atmospheric pressure (A3) 0.005 | Precipitation (A3) 0.004 | Atmospheric pressure (A2) 0.003 | Wind Speed (A3) 0.002 |
| GBM | Wind speed (A2) 0.008 | Atmospheric pressure (A3) 0.007 | Atmospheric pressure (A2) 0.007 | Wind speed (A3) 0.006 | Atmospheric pressure (A1) 0.005 |
| XGBoost | Wind speed (A2) 0.012 | Atmospheric pressure (A3) 0.008 | Atmospheric pressure (A1) 0.006 | Wind speed (A3) 0.006 | Atmospheric pressure (A2) 0.006 |
| LightGBM | Wind speed (A2) 0.011 | Atmospheric pressure (A3) 0.009 | Wind speed (A3) 0.007 | Atmospheric pressure (A2) 0.006 | Atmospheric pressure (A1) 0.006 |
| CatBoost | Wind speed (A2) 0.011 | Atmospheric pressure (A2) 0.008 | Wind speed (A3) 0.004 | Atmospheric pressure (A1) 0.004 | Atmospheric pressure (A3) 0.004 |

[1] RF: Random forest; ET: Extra tree; AdaBoost: Adaptive boosting; GBM: Gradient boosting machine; XGBoost: Extreme gradient boosting; LightGBM: Light gradient boosting machine; CatBoost: Categorical boosting. [2] A1: Approach 1—average value; A2: Approach 2—Maximum or minimum value; A3: Approach 3—Meteorological data at a specific time point. [3] The numbers below the meteorological variables represent the mean absolute SHAP value (unitless).

**Table 3.** Approach with the highest SHAP value for each meteorological variable using the proposed models.

| Model [1] | Meteorological Variable [2,3] | | | |
|---|---|---|---|---|
| | Air Temperature | Atmospheric Pressure | Precipitation | Wind Speed |
| RF | A3 (0.002) | A3 (0.006) | A3 (0.005) | A2 (0.008) |
| ET | A3 (0.002) | A2 (0.005) | A3 (0.004) | A2 (0.013) |
| AdaBoost | A1 (0.002) | A3 (0.005) | A3 (0.004) | A2 (0.019) |
| GBM | A3 (0.002) | A3 (0.007) | A3 (0.003) | A2 (0.008) |
| XGBoost | A3 (0.003) | A3 (0.008) | A3 (0.003) | A2 (0.012) |
| LightGBM | A1 (0.004) | A3 (0.009) | A3 (0.004) | A2 (0.011) |
| CatBoost | A2 (0.003) | A2 (0.008) | A3 (0.003) | A2 (0.011) |

[1] RF: Random forest; ET: Extra tree; AdaBoost: Adaptive boosting; GBM: Gradient boosting machine; XGBoost: Extreme gradient boosting; LightGBM: Light gradient boosting machine; CatBoost: Categorical boosting. [2] Approach (mean absolute SHAP value). [3] A1: Approach 1—Average value; A2: Approach 2—Maximum or minimum value; A3: Approach 3—Meteorological data at a specific time point.

### 4.4. Comparison of Model Performance

Five evaluation metrics, including MAE, MSE, RMSE, RMSLE, and $R^2$, were employed to compare the accuracy of the proposed fruit drop rate forecasting models.

The CatBoost outperformed the other models for all evaluation metrics, except for the MAE (Table 4). The hyperparameters tuned for the selected CatBoost model were iterations = 120, learning_rate = 0.3, depth = 8, random_strength = 0.7, min_data_in_leaf = 1, and eval_metric = 'RMSE'.

**Table 4.** Comparison of the performance of the proposed models on a test set using six evaluation metrics [1].

| Model [2] | Evaluation Metric | | | | |
|:---:|:---:|:---:|:---:|:---:|:---:|
| | **MAE** | **MSE** | **RMSE** | **RMSLE** | $R^2$ |
| RF | 0.0224 | 0.0020 | 0.0444 | 0.0395 | 0.3547 |
| ET | 0.0217 | **0.0018** | 0.0428 | 0.0381 | 0.3978 |
| AdaBoost | 0.0247 | 0.0021 | 0.0462 | 0.0411 | 0.3008 |
| GBM | **0.0209** | **0.0018** | 0.0428 | 0.0380 | 0.3995 |
| XGBoost | 0.0227 | 0.0019 | 0.0437 | 0.0389 | 0.3747 |
| LightGBM | 0.0215 | 0.0020 | 0.0442 | 0.0395 | 0.3578 |
| CatBoost | 0.0215 | **0.0018** | **0.0425** | **0.0378** | **0.4071** |

[1] The best result for each evaluation metric is highlighted in bold. [2] RF: Random forest; ET: Extra tree; AdaBoost: Adaptive boosting; GBM: Gradient boosting machine; XGBoost: Extreme gradient boosting; LightGBM: Light gradient boosting machine; CatBoost: Categorical boosting.

## 5. Discussion

This study analyzed the drop rate in apples due to typhoons using historical damage records from crop insurance for the period 2016–2021. Prediction models were used to determine the impact of typhoons, considering regional variations. Explanatory variables, including air temperature, atmospheric pressure, precipitation, and wind speed, were obtained from nationwide weather stations. Before establishing a model for accurately predicting typhoon damage based on meteorological data, it was necessary to carefully select the optimal measure of meteorological data that reflects typhoon damage. In this study, three measures were compared to determine the most suitable model input: the average value of the variable, the maximum or minimum value of the variable, and the value observed at a specific time point. The models' predictive performance was found to be better when applying these three approaches on a case-by-case basis for each meteorological variable (Table 4) compared to using only one approach for all variables (Tables A2–A4).

SHAP analysis was also conducted to assess the contribution of the predictor variables to the model's output. Furthermore, the proposed models were compared in terms of their predictive performance using various evaluation metrics. According to SHAP values, the maximum wind speed was found to have the highest contribution to the output in all models (Table 2). During the progression of the typhoon, it was confirmed that using the maximum or minimum value (Approach 2) or the value at a specific time point (Approach 3) of meteorological variables as input to the predictive model helped improve predictions, in contrast to using the average value (Approach 1).

Although CatBoost outperformed the other tested models in terms of all evaluation metrics except for MAE, it still produced an MAE of 0.0215 and RMSE of 0.0425. This prediction error was most likely due to the fact that the difference between apple varieties and windbreak facilities at individual orchards, which can affect the fruit drop both directly or indirectly, was not reflected in the prediction model. In addition, if the prediction model incorporates more information about the farms and includes more typhoon cases, it will be possible to generate more accurate predictions.

## 6. Conclusions

Research related to typhoon damage is important because typhoons are a major cause of premature fruit drop, leading to significant financial losses for fruit farms. Numerous studies have been conducted to examine the relationship between typhoon damage and various factors, but only a few exist that predict the expected damage. Providing predictive results before the typhoon arrives is important because it helps establish preventive measures in response to the impending typhoon. The WMO announced that the means to prevent disaster damage are forecasting and management capabilities. In the case of a tropical cyclone, it is said that issuing an alert 24 h before its occurrence can reduce the damage by 30 percent.

This study is the first attempt to predict the apple drop rate using meteorological data. Determining the extent of regional impact can be challenging, although typhoon characteristics (such as path and central pressure) or the Saffir–Simpson Hurricane Scale are suitable for describing the intensity of a typhoon. We analyzed the relationship between typhoon damage and meteorological variables using data obtained from weather stations installed across the country. Using meteorological data, it is possible for the proposed model to present the expected apple drop rate for each region when a typhoon occurs.

Typhoons usually affect Korea between July and October. This period overlaps with the harvest season for fruits, making it crucial not only for farmers as producers, but also for consumers and countries responsible for supply and demand policies. Therefore, the results derived from the fruit drop rate prediction model can be used as a basis for preparing a supply demand control strategy. The model estimates potential fruit crop losses caused by typhoons. Furthermore, these results are expected to be an effective way to reduce the damage caused by natural disasters in main crop-producing areas.

**Author Contributions:** Conceptualization, S.-H.L. and M.-S.K.; methodology, S.-H.C. and S.-Y.P.; software, S.-H.C. and S.-Y.P.; formal analysis, S.-H.C., S.-Y.P. and M.-S.K.; investigation, U.Y., B.L. and S.-H.L.; data curation, S.-H.C. and S.-Y.P.; writing—original draft preparation, S.-H.C., U.Y. and S.-Y.P.; writing—review and editing, S.-H.C., S.-Y.P., U.Y., B.L., S.-H.L. and M.-S.K.; supervision, S.-H.L. and M.-S.K.; project administration, S.-H.L.; funding acquisition, U.Y., B.L. and S.-H.L. All authors have read and agreed to the published version of the manuscript.

**Funding:** This work was supported by a grant from the Korea Institute of Planning and Evaluation for Technology in Food, Agriculture, Forestry and Fisheries (IPET) through the Open Field Smart Agriculture Technology Short-Term Advancement Program, funded by the Ministry of Agriculture, Food and Rural Affairs (MAFRA) (No. 322034-3), Republic of Korea.

**Institutional Review Board Statement:** Not applicable.

**Data Availability Statement:** Data sharing not applicable.

**Conflicts of Interest:** The authors declare no conflict of interest.

## Appendix A

Table A1 shows the period from occurrence to extinction of 24 impact typhoons from 2016 to 2021, out of the 155 typhoons that made landfall on the Korean Peninsula, as well as the period when typhoons actually caused damage to the main producing areas. It also includes information on the amount of damage and the apple drop rate caused by each typhoon. The amount of damage was collected by the Disaster Yearbook (2016–2021). The Saffir–Simpson Hurricane Scale (SSHS) is a classification criterion for hurricanes based on the intensity of sustained winds [42]. The SSHS classifies hurricanes as Categories 1–5, Tropical Storms (TS), and Tropical Depressions (TD). The higher the category number, the stronger the winds, followed by TS and TD.

Tables A2–A4 show the results of comparing the performance of models applied with three approaches—average value, maximum or minimum value, and observed value at a particular time point—for each meteorological variable.

**Table A1.** Period of existence and risk of the 24 impact typhoons from 2016 to 2021.

| Year | Typhoon | SSHS [1] | Period of Existence | Period of Risk | Amount of Damage [2] | Fruit Drop Rate [3] |
|------|---------|----------|---------------------|----------------|----------------------|---------------------|
| 2016 | Malakas | 4 | 09.13–09.20 | 09.17–09.21 | - | 0.000 ± 0.003 |
|      | Chaba | 5 | 09.28–10.06 | 10.03–10.06 | 214,465 | 0.016 ± 0.004 |
| 2017 | Nanmadol | TS | 07.02–07.05 | 07.02–07.05 | - | 0.000 ± 0.000 |
|      | Noru | 4 | 07.21–08.08 | 08.05–08.08 | - | 0.000 ± 0.000 |
|      | Talim | 4 | 09.09–09.18 | 09.16–09.18 | - | 0.000 ± 0.000 |
| 2018 | Prapiroon | 1 | 06.29–07.04 | 06.30–07.04 | 6416 | 0.000 ± 0.000 |
|      | Rumbia | TS | 08.15–08.18 | 08.15 | - | 0.000 ± 0.000 |
|      | Soulik | 3 | 08.16–08.25 | 08.22–08.25 | 9251 | 0.032 ± 0.060 |
|      | Trami | 5 | 09.21–10.01 | 09.29–10.01 | - | 0.000 ± 0.002 |
|      | Kong-Rey | 5 | 09.29–10.07 | 10.04–10.07 | 54,949 | 0.037 ± 0.068 |
| 2019 | Danas | TS | 07.16–07.20 | 07.19–07.20 | 3419 | 0.001 ± 0.005 |
|      | Francisco | 1 | 08.02–08.06 | 08.06 | 42 | 0.000 ± 0.002 |
|      | Lekima | 4 | 08.04–08.12 | 08.07–08.12 | - | 0.001 ± 0.005 |
|      | Krosa | 3 | 08.06–08.16 | 08.14–08.17 | 241 | 0.000 ± 0.000 |
|      | Lingling | 4 | 09.02–09.08 | 09.06–09.07 | 33,396 | 0.114 ± 0.095 |
|      | Tapah | 1 | 09.19–09.23 | 09.21–09.23 | 7977 | 0.070 ± 0.088 |
|      | Mitag | 2 | 09.28–10.03 | 10.01–10.04 | 167,704 | 0.024 ± 0.040 |
| 2020 | Jangmi | TS | 08.09–08.10 | 08.09–08.10 | - | 0.000 ± 0.000 |
|      | Bavi | 3 | 08.22–08.27 | 08.25–08.27 | 1122 | 0.017 ± 0.038 |
|      | Maysak | 4 | 08.28–09.03 | 09.01–09.03 | 221,419 | 0.108 ± 0.104 |
|      | Haishen | 4 | 09.01–09.07 | 09.04–09.07 | | 0.054 ± 0.076 |
| 2021 | Lupit | TS | 08.04–08.09 | 08.06–08.08 | - | 0.000 ± 0.001 |
|      | Omais | TS | 08.20–08.24 | 08.22–08.24 | 21,086 | 0.003 ± 0.020 |
|      | Chanthu | 5 | 09.07–09.18 | 09.15–09.17 | - | 0.000 ± 0.001 |

[1] SSHS: Saffir–Simpson Hurricane Scale. [2] Unit: 1 million won (KRW). [3] Mean ± standard deviation.

**Table A2.** Comparison of the models with meteorological variables from Approach 1 applied as model inputs (test set) [1].

| Model [2] | Evaluation Metric | | | | |
|-----------|-------------------|-----|------|-------|-----|
|           | **MAE** | **MSE** | **RMSE** | **RMSLE** | $R^2$ |
| RF | 0.027 | 0.003 | 0.050 | 0.045 | 0.190 |
| ET | 0.026 | **0.002** | **0.047** | **0.043** | **0.265** |
| AdaBoost | 0.028 | **0.002** | 0.049 | 0.044 | 0.209 |
| GBM | 0.026 | **0.002** | 0.048 | **0.043** | 0.256 |
| XGBoost | 0.026 | **0.002** | **0.047** | **0.043** | 0.264 |
| LightGBM | 0.027 | **0.002** | 0.049 | 0.044 | 0.214 |
| CatBoost | **0.025** | **0.002** | 0.048 | **0.043** | 0.237 |

[1] The best result for each evaluation metric is highlighted in bold. [2] RF: Random forest; ET: Extra tree; AdaBoost: Adaptive boosting; GBM: Gradient boosting machine; XGBoost: Extreme gradient boosting; LightGBM: Light gradient boosting machine; CatBoost: Categorical boosting.

**Table A3.** Comparison of the models with meteorological variables from Approach 2 applied as model inputs (test set) [1].

| Model [2] | Evaluation Metric | | | | |
|-----------|-------------------|-----|------|-------|-----|
|           | **MAE** | **MSE** | **RMSE** | **RMSLE** | $R^2$ |
| RF | 0.024 | **0.002** | 0.047 | 0.042 | 0.271 |
| ET | 0.024 | **0.002** | **0.046** | **0.041** | **0.301** |
| AdaBoost | 0.024 | **0.002** | 0.047 | 0.042 | 0.279 |
| GBM | **0.023** | **0.002** | 0.047 | 0.042 | 0.278 |
| XGBoost | 0.024 | **0.002** | 0.047 | 0.042 | 0.282 |
| LightGBM | 0.024 | **0.002** | 0.047 | 0.042 | 0.273 |
| CatBoost | 0.025 | **0.002** | 0.047 | 0.043 | 0.266 |

[1] The best result for each evaluation metric is highlighted in bold. [2] RF: Random forest; ET: Extra tree; AdaBoost: Adaptive boosting; GBM: Gradient boosting machine; XGBoost: Extreme gradient boosting; LightGBM: Light gradient boosting machine; CatBoost: Categorical boosting.

**Table A4.** Comparison of the models with meteorological variables from Approach 3 applied as model inputs (test set) [1].

| Model [2] | Evaluation Metric | | | | |
|:---:|:---:|:---:|:---:|:---:|:---:|
| | **MAE** | **MSE** | **RMSE** | **RMSLE** | $R^2$ |
| RF | 0.022 | **0.002** | 0.046 | 0.041 | 0.310 |
| ET | 0.022 | **0.002** | 0.046 | 0.041 | 0.295 |
| AdaBoost | 0.023 | **0.002** | 0.048 | 0.043 | 0.248 |
| GBM | 0.024 | **0.002** | 0.047 | 0.042 | 0.276 |
| XGBoost | 0.022 | **0.002** | 0.046 | 0.041 | 0.316 |
| LightGBM | 0.023 | **0.002** | 0.047 | 0.041 | 0.273 |
| CatBoost | **0.021** | **0.002** | **0.045** | **0.040** | **0.323** |

[1] The best result for each evaluation metric is highlighted in bold. [2] RF: Random forest; ET: Extra tree; AdaBoost: Adaptive boosting; GBM: Gradient boosting machine; XGBoost: Extreme gradient boosting; LightGBM: Light gradient boosting machine; CatBoost: Categorical boosting.

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
