# Peer review of "Analysis of the Fruit Drop Rate Caused by Typhoons Using Meteorological Data"

_agriculture, doi:10.3390/agriculture13091800_

Round 1
Reviewer 1 Report
The manuscript with the title “Analysis of the fruit drop rate caused by typhoons using meteorological data” predicted the fruit drop rates due to typhoons using historical claim records for crop and aided in estimating the potential loss to fruit crops.My specific comments are as follows:
1. Keywords: Apple should be revised as fruit.
2. Section 3.2: What is the basis for using temperature, atmospheric pressure, precipitation, and wind speed as explanatory variables?
3. Lines 364-370: How the three methods were selected?
4. Line 396,the 1,848 datasets were randomly divided into training and test sets at a 9:1 ratio, what is the basis for the division? According to the current situation, 7:3 and 6:4 are the universal division ratios.
5. Lack conclusion section.
Author Response
Dear Editor and Reviewers,
We would like to express our sincere gratitude to the reviewers and editor for their comments on our manuscript.
Taking into account the informative comments, we have identified the areas in which the manuscript could be enhanced.
Therefore, we are resubmitting a revised version of the manuscript along with our responses to the comments. The responses to the reviewers' comments will be submitted as a file.
Thank you for your consideration.

Author Response
Dear Editor and Reviewers,
We would like to express our sincere gratitude to the reviewers and editor for their comments on our manuscript.
Taking into account the informative comments, we have identified the areas in which the manuscript could be enhanced.
Additionally, We reviewed and corrected the overall grammatical error of the manuscript.
Therefore, we are resubmitting a revised version of the manuscript along with our responses to the comments. The responses to the reviewers' comments will be submitted as a file.
Thank you for your consideration.

Round 2
Reviewer 1 Report
- The conclusion needs to be refined.
Author Response

(The authors gave the same response as above.)

Reviewer 2 Report
The manuscript is acceptable now.
Some revisions are needed to improve the manuscript.
Author Response

(The authors gave the same response as above.)
